# Sequencing and analysis of 131 SARS-CoV-2 isolates in previously sampled and unsampled regions of Jordan from 2020 to 2023

Rame Khasawneh[1], Andrew W. Bartlow[2], Shirin Almharat[1], Abdullah Almuhasen[1], Ali Alhuniti[1], Lena Hajaya[1], Al Anood Al Habashneh[1], Osama Alshudefat[1], Mohammad Dojan[1], Abdelghani Tbakhi[3], Zeena Shaheen[4], Eiad Atwa[4], Colin PS Kruse[2], Osama Alsmadi[4]*, Samuel I. Koehler[2]*

1 Royal Medical Services Military Cancer Center, Amman, Jordan, 2 Los Alamos National Laboratory, Los Alamos, New Mexico, United States of America, 3 McMaster University Medical Centre, Ontario, Canada, 4 King Hussein Cancer Center, Amman, Jordan

* sikoehler@lanl.gov (SK); OA.12163@KHCC.JO (OA)

## Abstract

The Hashemite Kingdom of Jordan remains an understudied country for next generation sequencing analysis of SARS-CoV-2 genomes collected during the 2019 pandemic. Here we provide 131 additional reference genomes collected between 2020–2023 from SARS-CoV-2-positive patients across Jordan. Phylogenetic analysis supports existing pandemic narratives of changing clade dominance over time and adds genomes in novel Jordanian locations and timepoints to make Jordan SARS-CoV-2 databases more comprehensive. Samples from the less-sequenced cities of Ajloun, Jaresh, Karak, and Madaba identified previously unreported lineages while Amman, Irbid, and Zarqa have existing sequencing efforts bolstered. Despite many incomplete patient records and a relatively small sample size, we observe interesting symptom patterns that support existing global and Jordanian pandemic narratives. We note how in-country COVID-19 pandemic genomic studies showcase Jordan's efforts to expand next generation sequencing capabilities, especially through the leveraging of EDGE COVID-19, a bioinformatics platform for performing rapid, batched analysis of SARS-CoV-2 sequencing that streamlines sample processing prepared from a network of hospital locations.

## Introduction

Respiratory pathogens are a threat to global health security and cause high rates of morbidity and mortality [1,2]. These viruses are easily transmitted and have high mutation rates, making them especially challenging when developing therapeutics and tracking pathogen spread within and among communities. During a pandemic, sequencing has become an essential tool for tracking transmission patterns, viral

**Data availability statement:** All relevant data are within the paper and its Supporting information files.

**Funding:** This material is based upon work supported by the Cooperative Threat Reduction (CTR) Biological Threat Reduction Program (BTRP) under Award/Contract No. HDTRA1343361to Triad National Security, LLC, an operator of the Los Alamos National Laboratory under contract No. 89233218CNA000001 with the US Department of Energy. The funders had no role in study design, data collection and analysis, decision to publish, or preparation of the manuscript.

**Competing interests:** The authors have declared that no competing interests exist.

evolution, variant detection, and providing information to inform public health interventions. As the number of countries contributing sequences to public repositories grows, the power to understand pathogen evolution, transmission, and virulence also increases.

Many countries are adopting next generation sequencing (NGS) technology for pathogen detection and characterization to understand those impacting food security and human health [3,4]. The cost and steep learning curve of NGS makes it challenging to implement in biosurveillance programs. To alleviate the burden of establishing a sequencing facility, institutions receive support from cooperative engagement programs to learn sequencing and bioinformatics. These programs build trust between partner institutions and encourage data sharing, which is essential for pandemic preparedness and mitigation strategies [5].

As countries implement sequencing capabilities, success is limited by the hardware, software, and technical expertise needed to execute NGS analyses on tens to thousands of samples. The EDGE [6] bioinformatics platform was developed at Los Alamos National Laboratory to streamline many aspects of sequencing data analysis by reducing the complexity, cost, requisite knowledge, and tedium required for successful bioinformatics data processing. EDGE and the SARS-CoV-2 specific platform EDGE COVID-19 [7] are free, web-based applications implementing many common bioinformatics workflows. A variety of standard tools are integrated along with recommended default parameters to ensure successful bioinformatics results regardless of user experience. Furthermore, EDGE is deployed with a dedicated high performance computer backend, removing the costly and limiting requirement of server purchasing and maintenance. Here we demonstrate EDGE COVID-19 capability for handling the analysis of SARS-CoV-2 samples in an efficient, streamlined, and informative way.

Jordan was affected by the 2019 SARS-CoV-2 pandemic [8] in much the same way as the rest of the world. To avoid spreading the disease, various cultural interventions and travel restrictions were implemented. Despite Jordan's importance as a social and economic regional hub with many neighboring states (Iraq, Israel, Palestine, Saudi Arabia, Syria), relatively few whole genome sequences originate from the country, making pandemic tracking over time challenging. GISAID reports 1,645 (compared to 16,656,236 sequences globally) SARS-CoV-2 genome sequences derived from Jordan (as of 11 April 2024), which ranks 13/ 18 for total number of samples in GISAID for middle eastern countries (supplemental GISAID). Other SARS-CoV-2 next generation sequencing efforts in Jordan have explored the emergence of new variants [9], genetic diversity and geographic transmission patterns [10], correlations between country-implemented travel restrictions and transmission directionality [11], and associations between seasons and infection by specific viral lineages [12].

Jordan Royal Medical Services (JRMS) is the medical arm of Jordan Army forces and one of the largest health care systems in Jordan, covering more than 35% of Jordan's population through a country-wide network of referral specialized centers, hospitals (12), primary care centers, and clinics serving military and civilian patients. JRMS is a cornerstone of Jordan's fight against COVID-19, providing medical aid and expertise to support the nation's mitigation efforts with testing, treatment, and

vaccination services as well as establishing the first molecular lab in the Southern region of Aqaba. JRMS's unique hospital network and pandemic response role has allowed it to collect patient samples from previously unreported areas and, with its recent establishment of NGS capabilities, sequence and disseminate SARS-CoV-2 genomes. Demonstrated NGS ability caters to various medical needs and serves as vital surveillance sites for pathogens, allowing for early detection and monitoring of diseases in different regions of Jordan.

Countries that have advanced NGS infrastructure collaborate extensively across organizations to exchange expertise and leverage unique capabilities. As Jordan continues to develop sequencing efforts, establishing a NGS collaborative network to aggregate hardware, software, and technical expertise is crucial. King Hussein Cancer Center (KHCC), established in 1997 in Amman, Jordan, is a world-class cancer treatment and research institution in the Middle East with demonstrated NGS expertise. During the Covid-19 pandemic, KHCC maintained an active monitoring of SARS-CoV-2 among patients, many of whom are immunocompromised and considered highly at-risk for dramatic respiratory symptoms in addition to other clinical complications. PCR testing was routinely administered to symptomatic patients and employees to yield a diverse catalogue of SARS-CoV-2 patient samples. Co-analysis of pathogen samples between KHCC and JRMS offers an opportunity to demonstrate the utility of collaboration, notably between government and private organizations, to address national crises. This demonstration is further broadened by the inclusion of the Los Alamos National Laboratory Bioscience division (USA), an organization with historical involvement in major sequencing projects, to further inform and advance NGS efforts in Jordan and produce scientific insights.

Here, we provide 131 new whole genome sequences of SARS-CoV-2 derived from Jordan from 2020 to 2023 and explore how our results support narratives of shifting pathogen lineage dominance over time within-country.

## Methods

### Sample collection

Sampling occurred at local military hospitals (Ajloun, Amman, Irbid, Jarash, Karak, Madaba, Mafraq, Zarqa), in accordance with the U.S. Center for Disease Control and Prevention Interim Guidelines for Collecting, Handling, and Testing Clinical Specimens from Patients Under Investigation for 2019 Novel Coronavirus. Swab specimens were collected using iClean (BioMed Diagnostics; Singapore) nylon tipped swabs with plastic shaft and placed immediately into sterile tubes containing 3 ml of viral transport media with UTM-RT (W/O beads; Guangzhou, China) in addition to patient demographic information, including vaccination status, gender, age, geographic location, and clinical presentation. Specimens were packaged and transported to Princess Iman Research and Laboratory Sciences Center from all military hospitals throughout Jordan except for Aqaba, as these sample were tested in their newly established molecular lab, according to the current edition of the International Air Transport Association (IATA) Dangerous Goods Regulation, following shipping regulations for UN 3373 Biological Substance, Category B when sending potential SARS-CoV-2 specimens. Samples were stored at 2–8°C and shipped overnight to the testing facility on ice packs. Specimens were stored at 2–8°C for up to 48 hours after collection prior to RNA extraction. If RNA extraction could not be performed within 48 hours, samples were stored at −70°C.

Samples were also collected from the King Hussein Cancer Center (KHCC). Human nasopharyngeal swabs were used, as source for the viral RNA extraction. Participating patients and employees from KHCC were enrolled into this study. Upper respiratory specimen types included nasopharyngeal (NP) and oropharyngeal (OP) swab specimens collected by trained healthcare personnel. Swabs were placed immediately into a sterile transport tube containing 2–3 mL of viral transport medium (VTM), refrigerated, and shipped to the molecular diagnostics laboratory at KHCC for viral RNA extraction.

### Extraction and quality control of RNA

At JRMS, QIAamp Viral RNA Mini Kits were used to purify viral RNA for amplification, per manufacturer's instructions (Qiagen, Germany). Following extraction, RNA quality assessment was performed with a Qubit fluorometer (Qubit 4.0;

ThermoFisher; Waltham, MA). The RNA was diluted to the final standard input concentration of 10 ng/µl according to the manufacturer's instructions (Illumina Inc., MiSeq, CA). At KHCC, Qiagen manual QIAamp MinElute kit (Qiagen, Cat. # 57704) or the automated Qiasymphony viral pathogen extraction (Qiagen, Cat. # 937055) was used for RNA extraction, following the manufacturer's recommended protocols.

At KHCC, SARS-CoV-2 extracted RNA from PCR positive patients was used for PCR or kept frozen at −50°C for downstream NGS utilization. Patient RNA samples with relatively high titer levels (ct ≤ 24) were selected for sequencing to represent the disease pandemic in Jordan between the years 2020–2023. These samples were carefully selected to perform NGS deep sequencing to provide the high quality data necessary for downstream epidemiological and phylogenetic timeline profiling.

**Library generation, sequencing, and validation**

At JRMS, libraries of RNA samples were prepared for sequencing following the Illumina Covidseq workflow (Illumina Inc.) in accordance with the manufacturer's specifications. The extracted RNA was reverse transcribed to generate cDNA using AmpliSeq cDNA synthesis for Illumina (Illumina Inc., Cat. # 20022654). Each reverse transcription reaction requires 1–100 ng per pool of DNase-treated total RNA. The recommended input is 10 ng of RNA per pool. After total RNA was generated, cDNA was amplified in 2 pools according to the designed primers in AmpliSeq cDNA kit (Illumina Inc., Cat. # 20022654). The 2 pools were then recombined, fragmented, tagmented, and partially digested using FuPa Reagent. The tagmented amplicons were amplified once more with the addition of indexes to each sample using AmpliSeq CD Indexes for Illumina, DNA ligase, and switch solution. The indexed libraries were subjected to a post-tagmentation clean-up step using AM Pure XP beads with the Beckhman Coulter Life Sciences kit and 70% ethanol. Libraries were amplified using 50 µl Amplification Mix following the Equalizer workflow protocol (Illumina Inc.). A second clean-up was performed using Equalizer beads. An equal volume was transferred from each amplicon into a new microcentrifuge tube to be denatured and diluted as specified by the manufacturer workflow (Illumina Inc.). The libraries were quantified using Qubit 4.0 fluorometer by dsDNA HS Kit of Qubit High-Sensitivity Assay (Thermo Fisher Scientific, Cat. # Q32851). High Sensitivity DNA electrophoresis with Agilent 2100 bioanalyzer system assessed the size and quality of DNA over a range of sizes and concentrations using HS dsDNA Assay Kit. qPCR for Illumina by Biosystems 7500 using KAPA qPCR kit was employed to calculate the molarity of libraries then denatured by 0.1 fresh NaOH, which were then diluted to final loading concentration (10pM), as specified by the manufacturer. The resulting DNA libraries were sequenced on a MiSeq using reagent Kit v2 300-cycle.

At KHCC, SARS-CoV-2 Next Generation Sequencing was performed using either Illumina Covidseq workflow (Illumina Inc.), or QIAseq DIRECT SARS-CoV-2 Kit (Qiagen, Cat. # 333891) on an Illumina MiSeq sequencing platform. QIAseq DIRECT SARS-CoV-2 sequencing libraries were prepared according to the protocol provided by the manufacturer. The QIAseq DIRECT SARS-CoV-2 Kit was used following the enhanced rapid library preparation 4-hour workflow protocol, which enables a uniformly covered SARS-CoV-2 genome sequencing, and viral variants surveillance. The workflow was initiated with random-primed cDNA synthesis using 5 µl of the extracted viral RNA, with this input material determined positive by real-time PCR SARS-CoV-2 nasopharyngeal (NP) swab samples of ct cycles ≤ 24. Multiplexed primer pools were used to prepare two independent target-enriched PCR primers. SARS-CoV-2-enriched samples were next amplified and indexed with different sample indexes using unique dual indexes (UDIs) that enable accurate demultiplexing upon sequencing completion and analysis. For quality control, 10ul of each sample library was used for capillary electrophoresis on the Agilent TapeStation system. The resulting libraries were approximately 360–380 bp in size. All pooled libraries were diluted to 4nM, and from each library 5ul was taken to create an equimolar pool of all samples' libraries. A denaturation protocol was then applied, and finally the pooled library was sequenced using the Illumina MiSeq NGS platforms located in the sequencing core facility of King Hussein Cancer Center. The resulting FASTQ files generated from each NGS run, were exported for data analysis and visualization.

## Bioinformatics

For all bioinformatics analyses, default parameters were used unless otherwise stated. The EDGE COVID-19 [6] (EDGE –EU v2.4.0) web-based bioinformatics platform provides tools for both read-based and assembly-based analyses to enable rapid, automated, and standardized processes streamlined for GISAID submission. Sequencing run raw Illumina FASTQ files were uploaded and analyzed with "Pre-process" to quantify input reads, perform quality trimming, and remove human host reads (reference genome = GRCh38). The analysis "Reference-Based SARS-CoV-2 Genome Analysis" was performed using the *Severe acute respiratory syndrome coronavirus 2 isolate Wuhan-Hu-1, complete genome, (NC_045512.2)* as a reference to map reads, assemble a consensus genome, call variants, and identify viral lineage. Batch sample submission was implemented to enable identical analysis methods were performed across all 131 samples. Reference-assembled genomes containing less than 1,500 (>95% reference linear coverage) unknown nucleotide positions (n's) were uploaded to GISAID under EPI_SET ID EPI_SET_240515tu (GISAID SARS-CoV-2 genome repository is accessible at epicov.org under "search").

## Phylogenetic analysis

Reference-based consensus genome assemblies were uploaded to NextClade [13] and ran with default parameters to generate quality control information, assessing the following criteria as "good", "mediocre", or"bad": Missing Data, Mixed Sites, Private Mutations, Mutation Clusters, Frame Shifts, Stop Codons. Samples flagged "bad" for high abundances of missing bases (criteria = Missing Data, i.e., low linear coverage, threshold = >95%) and/or private mutations (criteria = Private Mutations, an estimate of sequencing errors) compared to the SARS-CoV-2 reference (NC_045512.2) were excluded from downstream analysis. Placement of filtered samples within the SARS-CoV-2 lineage tree was performed on EDGE COVID-19 [7] using UShER [14] v0.5.1. Tree branch order was manipulated in R [15] with the package ggtree [16] and metadata was visualized alongside the final tree using a custom script.

## Ethical clearance

The study was reviewed and received written approval from the LANL Human Subjects Research Review Board (HSRRB), the Committee for Clinical and Pharmaceutical Research and Professional Ethics (Jordan Royal Medical Services), and the Office of Human Research Protection Program at King Hussein Cancer Center (Protocol # 21 KHCC 31).

## Results

### Sampled patient data

Patient samples were collected from 2020–2023 and totaled 1 (1%), 20 (15%), 67 (51%), 30 (23%) by year, respectively (Fig 1B). Most samples were collected in 2022 and were from the city of Amman (96, 73%) with another large proportion collected in Irbid (11, 8%) (Fig 1B). Sixty-four male patients aged 1–78 and 48 female patients aged 6–70 were sampled, the majority of which were alive while a small fraction was deceased when sampled (2 males, 6 females) (Fig 1A). Four major symptoms were recorded in patients: *fever, joint pain, coughing, loss of smell*, and those exhibiting both fever and coughing were defined as having severe acute respiratory infections (SARI) as defined by WHO [17]. Regarding patient vaccination status, 79/ 131 (60%) of sampled patients had vaccination data recorded, with 14 (18%), 28 (35%), and 37 (47%) reporting no-vaccinations, Sinopharm, and Pfizer vaccines respectively (Fig 1C). Most patients from all vaccination categories reported fever symptoms, additionally most patients with vaccinations reported joint paint symptoms (Fig 1C). A third of patients with vaccination descriptions and positive SARS-CoV-2 results exhibited SARI symptoms (Fig 1C). Interestingly, unvaccinated patients had the largest proportion of individuals (5/14) exhibiting none of the reported symptoms (Fig 1C). Being vaccinated was not correlated with SARI symptom (having both fever and cough) occurrence (GLM: Chi-square = 3.75, d.f. = 2, P = 0.15. The average age of vaccinated patients was 37.2 while unvaccinated patients averaged 33.4.

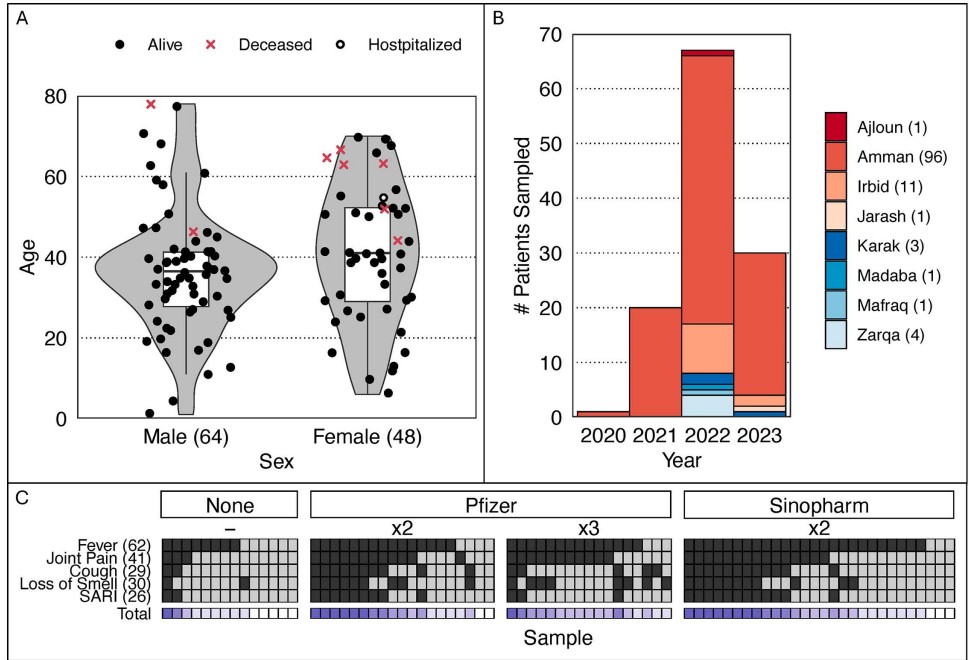

**Fig 1. Overview of collected patient metadata.** Violin plots with quartile ranges overlayed (panel A) are displayed by sex (x-axis) and patient age (y-axis), with mortality status described by point shape and color. Boxplots (panel B) break down patient sample collection by year (x-axis) and city (color), with total samples per city identified by parenthetical values and total samples by year indicated by y-axis sums. Patient symptoms (y axis) were recorded (panel C) for each sample (x axis), with SARI representing the co-occurrence of coughing and fever symptoms and *Total* values describing the total number of symptoms exhibited per patient and parenthetical values describing the total occurrence of each symptom across patients. Sample columns are grouped by vaccine received (Pfizer, Sinopharm, None=no vaccination) and doses administered (x2, x3).

## Sequence data quality control and reference alignment

In total, 131 samples were sequenced, yielding 33,864–5,880,710 raw reads and 31,799–5,729,474 cleaned reads after host removal and filtering, with most removed reads coming from human sequence removal (Fig 2).

Cleaned reads aligned to the standard reference *SARS-CoV-2 isolate Wuhan-Hu-1 complete genome* (NCBI accession NC_045512.2) produced linear base coverage above 75% for all samples despite a handful of samples reporting total reads-to-reference mapping rates below 60% due to high abundances of human host reads (S1 Fig). Average fold coverage of sample reads against the reference were all above the recommended depth of 100X, ranging from 143X – 22,228X (S1 Fig).

## Strain classification and phylogenetic analysis

Strain level classification was achieved for all samples (4 March 2023) and 99/ 131 were high enough quality (>95% linear coverage, suitable depth of coverage, minimal sequencing errors) for use in phylogenetic analyses, as defined by NextClade (Figs 3 and 4). Classification of sequenced samples fall into three chronological groupings. Samples dating December 2020 to December 2021 include many Pango lineages from BA, BQ, and XBB in addition to 20D, Alpha, and Delta strains which, notably, do not appear at any later date sampled. January 2022 – December 2022 is comprised almost exclusively of BA lineages proceeded by January – December 2023 samples being dominated by XBB strains (Figs 3 and 4).

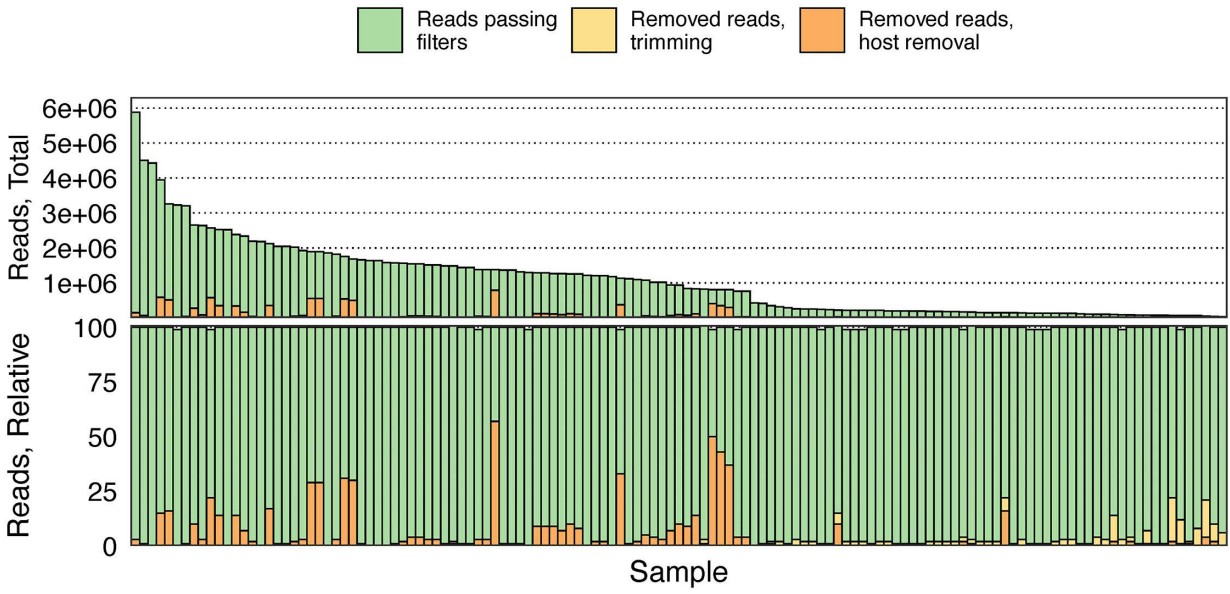

**Fig 2. Sequencing read quantities.** Total read quantities (row 1) and relative proportions (row 2) removed due to quality trimming (yellow) and removal of human-aligning sequences (orange). Individual bars (columns) define samples, organized from most to fewest reads yielded.

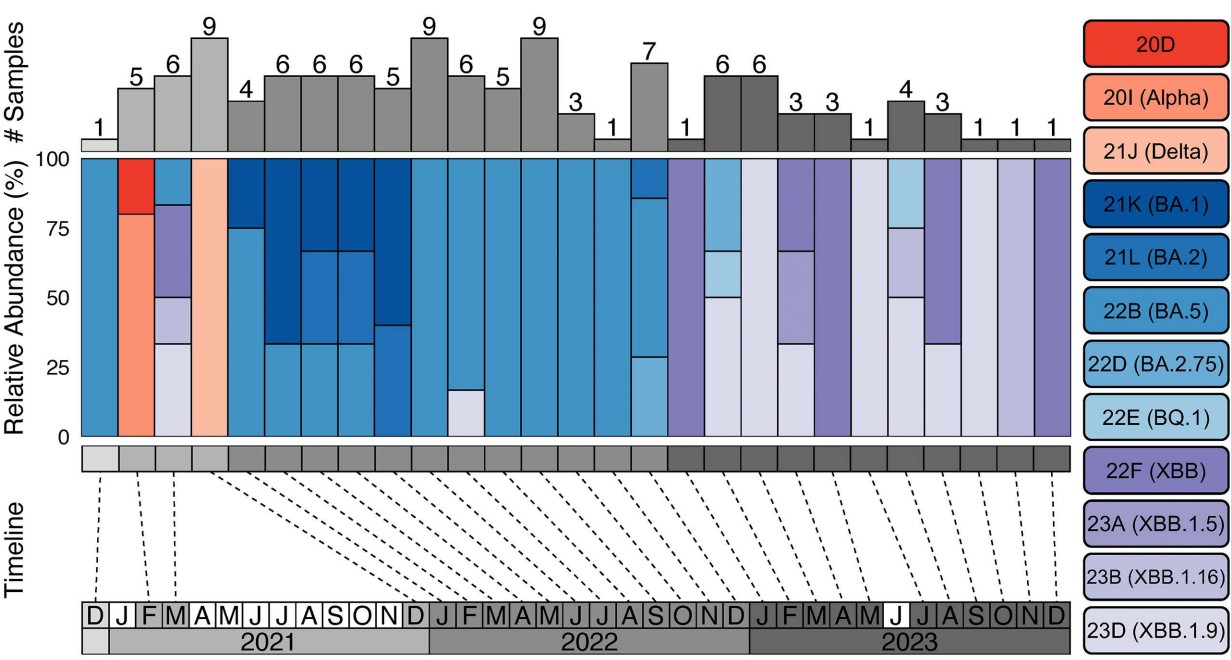

**Fig 3. SARS-CoV-2 Classification timeline.** Timeline displaying the quantity and strain classification (Nexstrain with Pango lineage in parentheses) of samples from December 2020 to December 2023. The bottom row displays collection year with months organized chronologically above. White months indicate no data samples were collected while progressively darker shades of gray define subsequent years. Dashed lines associate sample relative abundances (colored bars) with their collection times, and top row gray bars capture the total quantity of samples collected in specific months.

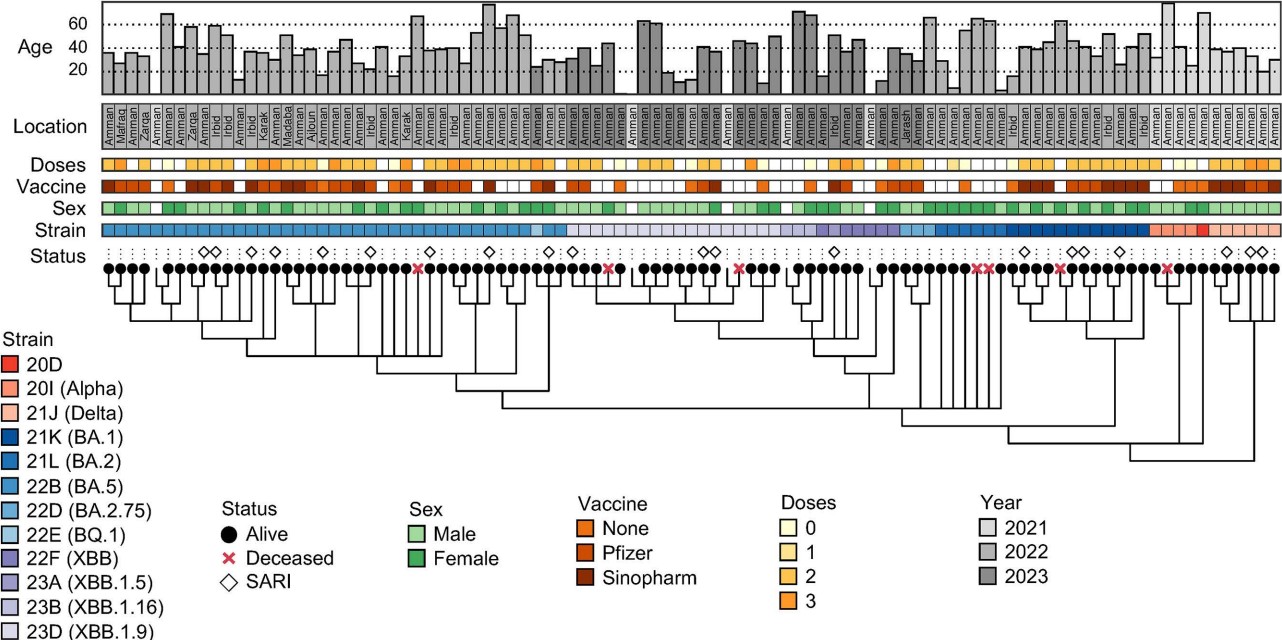

**Fig 4. Phylogenetic tree with patient metadata.** Phylogenetic tree of 99 sequenced covid samples passing NextClade quality filtering. In ascending order, metadata describes patient mortality (Status), UShER strain classification as described by Nexstrain with Pango lineage in parentheses (Strain), patient sex (Sex), patient vaccine type (Vaccine) and quantity of doses received (Doses), the year location the patient was sampled (Year), the location of the patient (Location), and the patient's age in years (Age). Location and age columns are colored by collection year.

UShER placement of study samples in the global Nextstrain [18] phylogeny built with ncov [19] yielded isolates classified in lineages 22B, 22D, 22F, 23B, and 23D (Fig 5). Lineages 22B, 22D, and 22F exist as parallel branches of the Nextstrain phylogeny (nextstrain.org/NextClade/sars-cov-2/21L), suggesting potential independent infection events while the observed 23B and 23D lineages are descended from 22F and may indicate further introductions. The strain 23C is descended from 22B and is not observed in our tree, suggesting that temporal restrictions around the time that 22D infections were prevalent may have prevented later 23C introductions. This pattern contrasts results for 22F classifications and their observed 23B and 23D descendants, which may suggest a failed attempt at preventing outside introduction events.

## Discussion

Here, we provide a significant contribution of Jordan-derived SARS-CoV-2 reference genomes, a majority of which (99/131) are high enough quality (complete or near-complete assemblies with few incorporated sequencing errors) for use in phylogenetic analyses (Fig 4). Samples rejected from phylogenetic analyses were primarily caused by missing and/ or ambiguous base calls at phylogenetically informative loci and do not necessarily indicate that the overall assembly is of poor quality.

While sample throughput remains insufficient for drawing broad claims about national pandemic patterns, strain classification results across sampled dates support and expand existing GISAID Jordan pandemic data. Most Jordanian GISAID samples are derived from Amman, leaving 468 out of 1645 (28%) from other cities. In the lesser-sampled cities of Ajloun, Jaresh, Karak, and Madaba, all previously published GISAID samples were derived from Royal Medical Services sequencing efforts. We add an additional sample to Ajloun's existing 13 and expand identified strains in the city from AY.6, AY.106, and B.1.617.2 to include BA.5.2. Jaresh saw one GISAID sample added to the existing 11, thereby expanding identified strains from A.5 and AY.106 to include BN.1. Karak's 10 total references had 3 added to expand an

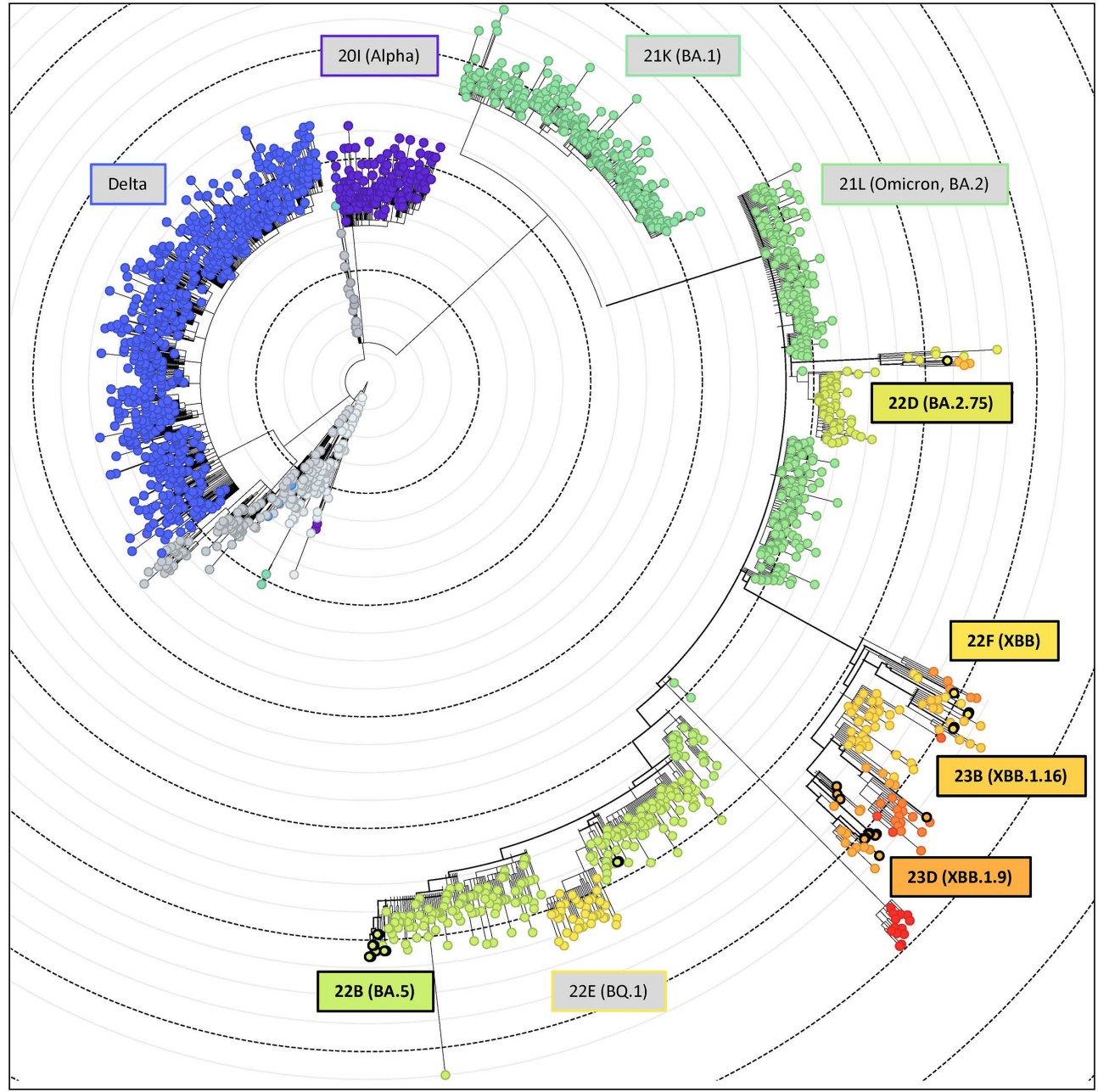

**Fig 5. UShER global phylogenetic tree.** UShER global phylogenetic tree visualized on Nexstrain displaying UShER-assigned Nexstrain clades. Black-outlined circles define individual USHeR samples from Jordan, with color-filled boxed defining present clades and gray boxes highlighting major lineages.

already diverse collection of identified strains (AY.106, AY.122, AY.93, B.1.617.2, BQ.1.1) to include BA.5.2, BA.5.2.34 while adding an additional BQ.1.1 sample. Madaba previously contained 10 samples, a single additional BA.5.2 sample was added to the list of identified strains (AY.106, AY.121, AY.73, B.1.617.2, BA.5.2). Mafraq has only 5 previous samples associated with it (3 from JRMS), despite this small quantity our additional sample was from strain BA.5.2.34 which was

previously identified in the city (AY.112, AY.106, B.1, BA.5.2.34). Zarqa has the third largest quantity of existing samples (83), collected by many organizations. While much strain diversity has been recorded from previous sequencing samples, our 4 additional Zarqa samples support previous findings by identifying more patients infected by variants BA.5.2 and BA.5.2.34. Irbid, the city with the second most samples in Jordan (192), exhibits a variety of identified strains and is supported by our 11 additional samples that show similar levels of diversity of previously present strains.

We can prepare for future pandemics by studying how outbreaks are managed by countries, with each presenting a unique set of circumstances and policy implementations. Insights are limited by the quality and robustness of pandemic data, especially pathogen dispersion mechanisms and locations with specific strain prevalences. This study expands knowledge of Jordanian city strain distributions during the Covid-19 pandemic. Crucially, this work demonstrates how a modest sequencing effort of 131 samples taken from under-sampled locations can fill out temporal pathogen strain distributions. The addition of 131 sequences to the existing 1645 GISAID samples will aid future global or regional analyses performed where the number of cities, strains, and timepoints increases statistical power.

While many months contain few or no sequencing data, when aggregated by year, there is a clear pattern of changing strain prevalence in our sequenced samples (Figs 3 and 4). Very few samples exist from 2021; however, they manage to cover a diversity of lineages not seen at any later year. It is also worth noting that this diversity of 2021 classification are all derived from samples from Amman (Fig 1). Results from 2022 and 2023 have more consistent sampling rates and clade classifications, with 2022 dominated by BA and BQ lineages and 2023 dominated by XBB lineages (Figs 3 and 4). Whereas 2021 samples are characterized as taxonomically diverse and all sampled from Amman, 2022 samples are unique in their geographic origin diversity with the same pattern holding albeit to a lesser extent for 2023 samples (Fig 1). Interestingly, months with the highest sampling rates (9 patients each in December 2021, June 2022, September 2022) have consistent Nexstrain classifications of 21J, 22B, and 22B respectively. Strain abundance patterns captured in samples from 2020–2022 support global patterns, with Alpha and Delta lineages representing the majority of reported infections until January 2022 when BA lineages come to dominate [20]. The 2021 publication by Sallam and Mahafzah [10] analyzed 579 GISAID SARS-CoV-2 samples collected from across Jordan (specific locations not reported) from March 2020 – January 2021 describe Jordan lineage patterns leading up to our sampling dates that support reported findings (B lineages dominating in late 2020) as well as global trends during this period, as previously described. CDC reporting through May 2023 reveal a decline in BA lineages and an increasing dominance of XBB lineages [21]. Our results reflect more-recent reported CDC findings as well, with later 2022 – early 2023 samples producing a transition from BA to XBB lineage samples [21]. While temporal lineage classification results are not expansive enough to draw conclusions about Jordan SARS-CoV-2 evolution dynamics, they bolster existing global and regional pandemic narratives.

Jordan's largest COVID-19 published sequencing effort, performed by Parker *et al.*, reported that patterns of transmission in-country were similar to those found elsewhere in Africa and Europe, with regional connections and land-based travel driving strain introduction and subsequent infection risk profiles [11]. They note that while Jordan's NGS infrastructure remains inadequate for monitoring transmission dynamics in-country, continued efforts to sequence the virus in-country may provide a sampling perspective that can help in informing epidemic dynamics and governmental policy implementations [11]. Previous results published by JRMS [22] saw association between region and lineage dominance, emphasizing the need for greater sequencing efforts to accurately capture diversity and spread mechanisms within country. Sallam and Mahafzah [10] demonstrate the utility of in-country sequencing projects, showing that accurate estimates of variance frequency can be modeled to more accurately explore transmission timelines, claiming that the actual introduction of lineage B.1.1.7 was four weeks earlier than previously reported [10]. Another publication from Jordan in 2021 identified the emergence of a previously-unreported variant [9]. NGS capabilities allow cities, countries, and regions to have a more accurate understanding of local pandemic evolution for informing public policy to reduce mortality and economic losses. The greater these capabilities are, the more these positive consequences will be magnified. Our contribution of in-country SARS-CoV-2 sequences, especially from under-sampled cities, will give these individuals greater

resources to draw from when exploring lessons from the COVID-19 pandemic and preparing for inevitable future outbreak events.

Across all patient vaccination categories (unvaccinated, Sinopharm with 2 doses, Pfizer with 2 doses, Pfizer with 3 doses) fever remained the most prevalent reported symptom, being reported in the majority in each category. Most unvaccinated patients display one or zero symptoms (11/14), which stands in stark contrast with the other vaccination categories where symptom reporting rates were much higher (Fig 1). Qaqish et al. report that national vaccination efforts in Jordan beginning January 2021 were not accepted by the majority of the population (~70%), including "knowledgeable" groups such as college students [23], which may explain the lack of symptoms as people with vaccine distrust may have skewed perceptions of their health. Of the 11 asymptomatic, unvaccinated patients, 3 were minors (ages 1, 11, and 14). The reduced efficacy of the Sinopharm vaccine compared to Pfizer has been previously observed [24]. Importantly, the disproportionate sample sizes of the vaccinated and unvaccinated populations indicates a likely sample bias wherein vaccinated patients are intrinsically more likely to seek medical care when symptomatic/sick. These results contrast with previous findings, where 45.5% (97/213) of sampled patients were unvaccinated [22]. Such discrepancies highlight the need for increased sequencing to inform potential metadata patterns. Identifying and clarifying patient data patterns can benefit public safety efforts as well. For example, confusion surrounding the safety of the vaccine has been reported in Jordan, with one study claiming that Jordanians exhibit proactive tendencies towards the disease [25]. Another analysis of cancer patient public opinion found that 54.7% (n = 441) of those questioned believed that the vaccine was unsafe, with 49% claiming that their oncologist had not educated them on vaccination benefits [26]. Pfizer 3-dose patients rarely exhibited both SARI symptoms of coughing and fever, instead usually only reporting fever but no cough. While this last observation may support the utility of additional booster doses, it is worth noting that the Pfizer 3-dose patient cohort contained all major covid lineages reported here.

Deceased patients similarly fall across all major lineages reported (Fig 4). All deceased patients in our study are older than 40 years old and majority women, a finding consistent with Jordanian polling by Mahmud et al. from June-September 2021 which found that older individuals (ages 40−64), particularly women, reported being less likely to have positive intention of being vaccinated [27], which can enhance fatality risk. Symptom patterns show some resemblance to other work published in Jordan, where 61 SARS-CoV-2 positive children were observed to exhibit symptoms in low proportions: 27 (44%) asymptomatic, 11 (18%) with fever, and 7 (11.5%) with cough [28].

Sample size limitations and sampling location bias prevent us from being able to model broad patterns across Jordan. Despite this limitation, interesting observations can be made on global COVID-19 pandemic properties compared to the SARS-CoV-2 lineages captured at specific dates in this study. Samples placed in a global tree by UShER fell within five lineages: 22D, 22F, 23B, 23D, and 22B which align chronologically with global dominant strain trends [20]. The absence of earlier strains in our data may reflect the timing of our sampling, which primarily occurred in 2022 and 2023. Jordan's early response to SARS-CoV-2 from mid-March – August 2020 involved the closure of most industry, curfews, and social distancing [29] and may have contributed to a lower prevalence of early-variants due to effective limiting of pathogen spread. Interestingly, other large lineages from similar dates are absent, notably 21K (BA.1), 21L (Omicron BA.2) and 22E (BQ.1) (Fig 5). While 21L samples were found in our analysis (7 samples, Fig 5), they were rejected by UShER due to insufficient quality emphasizing the need for greater sequencing breadth to properly capture pandemic diversity. Similarly, patients with Omicron were found in Jordan beginning October 2021 and by January 2022 accounted for 55% of new cases [23] as observed in our results beginning January 2022 when sample collection becomes monthly and are identified as Omicron (Figure 3).

The presence of 22B through 23D lineages could have, in real time, indicated ongoing lineage introduction events to clarify and support policy actions for preventing further viral spread. March 2020 witnessed Jordan's initial major response to the pandemic when policies to minimize travel and human-to-human contact were implemented [29]. In September 2020 commercial trucks were allowed to resume routes and children returned to the classroom [23], likely increasing

pathogen transmission. In December 2020 the Jordan Ministry of Health began their vaccination campaign against SARS-CoV-2 [30], which initially experienced low civilian acceptance [27], allowing viral transmission to continue. Our data, collected mostly in 2022 and 2023, suggest that public health decisions did not prevent intra/inter-country viral transmission in the third and fourth years of the pandemic, as strains common in the rest of the world were found to also be circulating in Jordan at the same time (Figure 5). Additionally, the presence of common strains in smaller Jordanian cities suggest that travel was occurring to these regions and was also enabling viral transmission. By highlighting these patterns, we hope to inform retrospective identification of likely transmission mechanisms which may be relevant during the next pandemic.

As many countries begin adopting next generation sequencing technologies, high-profile use cases can demonstrate the value and efficacy of leveraging genomic techniques. The SARS-CoV-2 pandemic demanded the production of much sequencing data, providing countless companies, cities, and governments the opportunity to demonstrate their NGS and bioinformatics capabilities for informing life-saving public policy decisions. SARS-CoV-2 has a small genome size, readily observable symptoms, and low genomic complexity, making it an ideal virus for teaching sequencing and bioinformatics. With the EDGE COVID-19 platform managing bioinformatics analysis, a lab can quickly adopt NGS testing capabilities if provided appropriate lab space, a sequencing machine, and an internet connection. The deployment of EDGE COVID-19, a pared-down version of the core EDGE platform, contains analysis capabilities streamlined for SARS-CoV-2. Having a website dedicated to SARS-CoV-2 bioinformatics reduces the burden of learning an entire new software suite and enabled us to efficiently understand our analysis options to begin data processing and interpretation. Many resource-limited laboratories have access to computers suitable for web browsing but not intensive bioinformatics analysis. EDGE alleviates computer hardware limitations by providing access to dedicated servers for large computational tasks so that research and analysis can be performed on simple computer setups. Analysis is not the final step of pathogen science, but it is often a limiting step. The EDGE platform reduces analysis bottlenecks but is not a comprehensive solution for interpreting pathogen studies. EDGE COVID-19 facilitated strain identification of our samples, then further literature review and meta data-supplemented analysis were required to contextualize samples in the broader Covid-19 pandemic context. We predict that similar pathogen-specific instances of EDGE will be readily deployed for future pandemics as they arise, enabling similar rapid response efforts as seen with Covid-19.

The collection and sequencing of COVID-19 samples in this study were pivotal for identifying the genomic characteristics and evolutionary dynamics of the virus within Jordanian communities. By analyzing these samples, we aimed to track the spread of different viral strains, identify potential variants of concern, and assess the effectiveness of public health interventions. This data driven approach is essential for informing targeted containment strategies, optimizing diagnostic protocols, and guiding the development of therapeutics and vaccines protocols, ultimately aiding in the ongoing efforts to control the spread of COVID-19.

In order to draw broad conclusions about pandemic patterns, a significant portion of infected cases must be recorded and sequenced. In the absence of mass-scale sequencing, carefully executed experiments may aim to capture a representative sub-sample of larger populations to statistically predict patterns. Here, our sample population is neither large nor sufficiently representative for making broad claims. Despite these significant limitations, we believe our data set is valuable to report in a severely under-studied region that may still be useful in identifying highly general pandemic patterns in Jordan. We recognize that our dataset is heavily skewed by time and location, and therefore its primary utility is in verifying the presence of specific strains in particular regions over defined date ranges. As additional studies emerge from Jordan and surrounding countries, we will have presented our data in such a way that it may be readily integrated into larger analyses and conclusions. Being able to efficiently sequence and analyze genomes bridges the gap between sample collection and actionable results that can be provided to governmental entities for guiding outbreak control strategies. For labs with emerging NGS capabilities, we demonstrate that leveraging collaboration and free online resources can greatly facilitate these efforts.

 

## Supporting information

**S1 Fig. Sequencing read data quality metrics.** Statistics describing alignment performance of cleaned reads against the reference SARS-CoV-2 Wuhan-Hu-1 complete genome. Induvial dots describe the performance of samples for percentage of total cleaned reads mapped to the reference (left), the percent of reference bases covered by one or more sample reads (middle), and the average fold coverage of reads mapped against the reference (right).
(TIFF)

**S1 Table. Sampled patient metadata.** Table describing information from sampled SARS-coV-2-positive patients. *Sample_Name*, Patient sample name used for EDGE analysis and GISAID submission; *Collection_YYYY.MM.DD*, The year, month, and date that samples were collected from patients; *Age*, Patient age; *Country*, Patient country of origin; *Location*, Patient city of origin; *Sex*, Patient biological sex; *Host_Status*, Mortality status of patient; *Vaccine*, The type of vaccination the patient has recieved; *n_Doses*, The number of vaccine doses the patient has recieved; *Category_Symptom*, Patient rating of experienced symptoms; *Symptom_Fever*, Presence/absence of fever; *Symptom_Pain_Joint*, Presence/absence of joint pain; *Symptom_Loss_Smell*, Presence/absence of loss of smell; *Symptom_Cough*, Presence/absence of coughing; *Symptom_Cold*, Presence/absence of cold; *Symptom_Loss_Taste*, Presence/absence of taste loss; *Symptom_Headache*, Presence/absence of headache; *Symptom_Breathing*, Presence/absence of breathing difficulty; *Symptom_Pain_Back*, Presence/absence of back pain; *Symptom_Pain_Chest*, Presence/absence of chest pain; *Symptom_Chills*, Presence/absence of chills; *CT*, qPCR cycle threshold of detection; *Reads_Raw*, Quantity of raw sequencing reads; *Reads_Clean*, Quantity of sequencing reads passing quality filtering; *cntg_n*, The quantity of assembled contigs; *cntg_max*, Maximum contig size in base pairs; *assembly_bp*, Total base pairs in assembly; *clade_Nextstrain*, Nextstrain clade assignment of genome assembly.
(XLSX)

## Acknowledgments

We would like to thank the Defense Threat Reduction Agency – Cooperative Threat Reduction Program for facilitating collaboration and communication between partner organizations. We also thank Cheryl Gleasner and the rest of the LANL Sequencing team for the NGS guidance provided to Royal Medical Services throughout sequencing efforts. We thank the leadership at the King Hussein Cancer Center (KHCC) for their support and financial backing of this study (Protocol 21 KHCC 31).

## Author contributions

**Conceptualization:** Rame Khasawneh, Andrew W. Bartlow, Shirin Almharat, Osama Alsmadi, Samuel I Koehler.

**Data curation:** Rame Khasawneh, Shirin Almharat, Abdullah Almuhasen, Lena Hajaya, Colin PS Kruse, Osama Alsmadi, Samuel I Koehler.

**Formal analysis:** Rame Khasawneh, Andrew W. Bartlow, Shirin Almharat, Abdullah Almuhasen, Ali Alhuniti, Lena Hajaya, Al Anood Al Habashneh, Osama Alshudefat, Mohammad Dojan, Abdelghani Tbakhi, Zeena Shaheen, Eiad Atwa, Colin PS Kruse, Osama Alsmadi, Samuel I Koehler.

**Funding acquisition:** Rame Khasawneh, Andrew W. Bartlow, Osama Alsmadi.

**Investigation:** Rame Khasawneh, Andrew W. Bartlow, Shirin Almharat, Osama Alsmadi, Samuel I Koehler.

**Methodology:** Rame Khasawneh, Andrew W. Bartlow, Shirin Almharat, Osama Alsmadi, Samuel I Koehler.

**Project administration:** Rame Khasawneh, Andrew W. Bartlow, Samuel I Koehler.

**Resources:** Rame Khasawneh, Andrew W. Bartlow, Osama Alsmadi.

**Software:** Andrew W. Bartlow, Abdullah Almuhasen, Lena Hajaya, Colin PS Kruse, Samuel I Koehler.

**Supervision:** Rame Khasawneh, Andrew W. Bartlow, Shirin Almharat, Osama Alsmadi, Samuel I Koehler.

**Validation:** Colin PS Kruse, Samuel I Koehler.

**Visualization:** Samuel I Koehler.

**Writing – original draft:** Rame Khasawneh, Andrew W. Bartlow, Colin PS Kruse, Osama Alsmadi, Samuel I Koehler.

**Writing – review & editing:** Rame Khasawneh, Andrew W. Bartlow, Colin PS Kruse, Osama Alsmadi, Samuel I Koehler.

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
