## [Decision Letter · Decision Letter 0]

16 Sep 2024

Dear Dr. Koehler,

Thank you for submitting your manuscript to PLOS ONE. After careful consideration, we feel that it has merit but does not fully meet PLOS ONE’s publication criteria as it currently stands. Therefore, we invite you to submit a revised version of the manuscript that addresses the points raised during the review process.

We look forward to receiving your revised manuscript.

Kind regards,

Nihad A.M Al-Rashedi

Academic Editor

PLOS ONE

Journal requirements: 1. When submitting your revision, we need you to address these additional requirements. Please ensure that your manuscript meets PLOS ONE's style requirements, including those for file naming. The PLOS ONE style templates can be found at https://journals.plos.org/plosone/s/file?id=wjVg/PLOSOne_formatting_sample_main_body.pdf and https://journals.plos.org/plosone/s/file?id=ba62/PLOSOne_formatting_sample_title_authors_affiliations.pdf. 2. We note that the grant information you provided in the ‘Funding Information’ and ‘Financial Disclosure’ sections do not match.  When you resubmit, please ensure that you provide the correct grant numbers for the awards you received for your study in the ‘Funding Information’ section. 3. Thank you for stating the following financial disclosure:  [This material is based upon work supported by the Cooperative Threat Reduction (CTR) Biological Threat Reduction Program (BTRP) under Award/Contract No. HDTRA1343361to Triad National Security, LLC, an operator of the Los Alamos National Laboratory under contract No. 89233218CNA000001 with the US Department of Energy.].  Please state what role the funders took in the study.  If the funders had no role, please state: ""The funders had no role in study design, data collection and analysis, decision to publish, or preparation of the manuscript."" If this statement is not correct you must amend it as needed. Please include this amended Role of Funder statement in your cover letter; we will change the online submission form on your behalf. 4. Thank you for stating the following in the Acknowledgments Section of your manuscript: [This material is based upon work supported by the Cooperative Threat Reduction (CTR) Biological Threat Reduction Program (BTRP) under Award/Contract No. HDTRA1343361 to Triad National Security, LLC, an operator of the Los Alamos National Laboratory under contract No. 89233218CNA000001 with the US Department of Energy. ]We note that you have provided funding information that is not currently declared in your Funding Statement. However, funding information should not appear in the Acknowledgments section or other areas of your manuscript. We will only publish funding information present in the Funding Statement section of the online submission form. Please remove any funding-related text from the manuscript and let us know how you would like to update your Funding Statement. Currently, your Funding Statement reads as follows:   [This material is based upon work supported by the Cooperative Threat Reduction (CTR) Biological Threat Reduction Program (BTRP) under Award/Contract No. HDTRA1343361to Triad National Security, LLC, an operator of the Los Alamos National Laboratory under contract No. 89233218CNA000001 with the US Department of Energy.]. Please include your amended statements within your cover letter; we will change the online submission form on your behalf. 5. Please provide a complete Data Availability Statement in the submission form, ensuring you include all necessary access information or a reason for why you are unable to make your data freely accessible. If your research concerns only data provided within your submission, please write "All data are in the manuscript and/or supporting information files" as your Data Availability Statement. 6. Please include your full ethics statement in the ‘Methods’ section of your manuscript file. In your statement, please include the full name of the IRB or ethics committee who approved or waived your study, as well as whether or not you obtained informed written or verbal consent. If consent was waived for your study, please include this information in your statement as well. 7. We are unable to open your Supporting Information file [Table S1.tsv]. Please kindly revise as necessary and re-upload.

Reviewers' comments:

Reviewer's Responses to Questions

**Comments to the Author**

1. Is the manuscript technically sound, and do the data support the conclusions?

Reviewer #1: Yes

Reviewer #2: Yes

2. Has the statistical analysis been performed appropriately and rigorously?

Reviewer #1: Yes

Reviewer #2: Yes

3. Have the authors made all data underlying the findings in their manuscript fully available?

Reviewer #1: Yes

Reviewer #2: Yes

4. Is the manuscript presented in an intelligible fashion and written in standard English?

Reviewer #1: Yes

Reviewer #2: Yes

Reviewer #1: This manuscript presents an analysis of 131 SARS-CoV-2 genomes sequenced from samples collected in Jordan between 2020-2023. The study provides valuable genomic data from an understudied region and demonstrates the utility of cooperative partnerships in developing sequencing and bioinformatics capabilities. Overall, the work contributes useful information on SARS-CoV-2 lineages circulating in Jordan over time. However, there are some areas that could be improved to strengthen the manuscript:

Major comments:

1. The sample size is relatively small (131 genomes) compared to many published SARS-CoV-2 genomic epidemiology studies. The authors should more clearly acknowledge this limitation and discuss how it may impact the conclusions that can be drawn

2. The phylogenetic analysis and discussion of lineage dynamics over time could be expanded. More detail on how the observed patterns compare to regional and global trends would provide valuable context.

3. The statistical analysis of symptoms and vaccination status (lines 265-271) is limited by small sample sizes in some groups. The authors should be more cautious in drawing conclusions from these analyses and clearly state the limitations.

4. The discussion of the utility of the EDGE-COVID-19 platform feels somewhat disconnected from the rest of the manuscript. This section could be condensed and better integrated with the main findings.

Minor comments:

5. Lines 79-91: The background on Jordan's healthcare system and involvement in COVID-19 response is useful but feels somewhat out of place in the introduction. Consider moving some of this information to the methods or discussion.

6. More details on the quality control criteria used to exclude samples from phylogenetic analysis would be helpful.

7. Figure 1: The color scheme in panel C is difficult to interpret. Consider using a more color-blind friendly palette

In summary, while this manuscript provides valuable genomic data and insights from an understudied region, addressing these comments would help to strengthen the analysis and presentation of the findings. With appropriate revisions, this work could make a solid contribution to our understanding of SARS-CoV-2 genomic epidemiology in the Middle East.

Reviewer #2: This manuscript reports sequencing and analyses of 99 COVID-positive samples from Jordan. However, I don't think it is adequate to be published with the current write-up, as:

1. The authors are trying to showcase the outcome is the collaborative research between two countries which enable NGS-based sequencing and bioinformatic analyses. Publication in research journal should merit research outcome i.e. the results and discussion but not the collaborative efforts. This is not a platform for this purpose. Unless the authors willingly remove credits on the collaboration and emphasize only on the results and discussion, this manuscript could then be considered for publication in a scientific journal.

2. The sample size is too small to represent a country. This manuscript should go for a short communication or brief report.

Other minor comments:

1. Should provide the GISAID ID i.e. uploaded and accepted sequences in the manuscript, in stead of the submission form to GISAID (suppl file).

2. The data analyses should be specific. It mentioned that the sequence reads were mapped to the reference genome. But the subsequent description revealed that the genomic sequences used are assembled (de novo?).

3. Do you mean some of the assembled genomes only cover 65% of the reference genome, or reads mapping rate is 65%? I prefer only assembled genome sequences at complete and near complete to be used for subsequent analyses to avoid loss of data and inaccurate assignment of lineage and interpretation of prevalence of strains.

**Do you want your identity to be public for this peer review?** For information about this choice, including consent withdrawal, please see our Privacy Policy

Reviewer #1: No

Reviewer #2: No

---

## [Author Response · Author response to Decision Letter 1]

7 Nov 2024

Response to Reviewers, Reviewer #1

Major comments:

1. The sample size is relatively small (131 genomes) compared to many published SARS-CoV-2 genomic epidemiology studies. The authors should more clearly acknowledge this limitation and discuss how it may impact the conclusions that can be drawn

Thank you for bringing this to our attention. We agree that our sample size is relatively small and have revised many sections (Abstract, Introduction, Discussion) to properly contextualize our framing and interpretation of the results.

2. The phylogenetic analysis and discussion of lineage dynamics over time could be expanded. More detail on how the observed patterns compare to regional and global trends would provide valuable context.

Thank you for bringing this to our attention. We have expanded a paragraph in the Discission expanded to more thoroughly compare our findings to existing global and Jordan covid-19 pandemic phylogenetic reporting.

3. The statistical analysis of symptoms and vaccination status (lines 265-271) is limited by small sample sizes in some groups. The authors should be more cautious in drawing conclusions from these analyses and clearly state the limitations.

Thank you for pointing this out. We decided to remove this analysis from the manuscript to avoid over stating the importance of these results based on small sample sizes. We still describe the vaccination patterns qualitatively. We also left in the logistic regression of SARI symptoms and vaccine type. This is a

4. The discussion of the utility of the EDGE-COVID-19 platform feels somewhat disconnected from the rest of the manuscript. This section could be condensed and better integrated with the main findings.

Thank you for identifying this disconnection. We have revised the manuscript to better contextualize EDGE in relation to the development of new NGS capabilities and how the platform can leverage this objective. Many sections have been re-written to incorporate this.

Minor comments:

5. Lines 79-91: The background on Jordan's healthcare system and involvement in COVID-19 response is useful but feels somewhat out of place in the introduction. Consider moving some of this information to the methods or discussion.

Thank you for this helpful comment. This section has been moved from the introduction to the end of the discussion.

6. More details on the quality control criteria used to exclude samples from phylogenetic analysis would be helpful.

We agree that the quality control criteria was lacking. The section “Phylogenetics” has been expanded to better describe this.

7. Figure 1: The color scheme in panel C is difficult to interpret. Consider using a more color-blind friendly palette.

Thank you for pointing this out. Figure 1.C has been revised to be more color-blind friendly.

Response to Reviewers, Reviewer #2:

Major comments:

1. The authors are trying to showcase the outcome is the collaborative research between two countries which enable NGS-based sequencing and bioinformatic analyses. Publication in research journal should merit research outcome i.e. the results and discussion but not the collaborative efforts. This is not a platform for this purpose. Unless the authors willingly remove credits on the collaboration and emphasize only on the results and discussion, this manuscript could then be considered for publication in a scientific journal.

We thank the reviewer for bringing this to our attention. Our primary research focus here was to report on the sequencing and analysis of 131 covid sequences produced in Jordan. This quantity of samples represents a significant proportion for Jordan SARS-CoV-2 genomes and includes many samples collected from novel regions at dates not previously reported. To this end, we strongly believe that this work constitutes scientific outcomes worthy of publication with PLOS ONE. We agree with the reviewer that the focus of publication is not to showcase collaboration, and to this end we have significantly reduced emphasis on collaboration. Limited discussion of collaboration remains in the text, as we believe that demonstrating collaboration as a means towards developing NGS capability in a country with minimal sequencing productivity represents an important milestone worth highlighting.

2. The sample size is too small to represent a country. This manuscript should go for a short communication or brief report.

Thank you for identifying this limitation. A similar comment was made up by Reviewer #1 and has been addressed in detail, see responses to Reviewer #1 comments 1-3. We believe that while small, our samples and reporting represent an important contribution to the SARS-CoV-2 reporting for a country that is developing its NGS capabilities and has relatively few published genomic resources available in databases. This work represents a major contribution to the total available SARS-CoV-2 genome sequences collected from within Jordan, notably at many locations and dates that have not yet been reported.

Minor comments:

1. Should provide the GISAID ID i.e. uploaded and accepted sequences in the manuscript, instead of the submission form to GISAID (suppl file).

The section “Bioinformatics” was revised to describe how uploaded and accepted sequences may be accessed on GISAID.

2. The data analyses should be specific. It mentioned that the sequence reads were mapped to the reference genome. But the subsequent description revealed that the genomic sequences used are assembled (de novo?).

Thank you for this comment. The data analysis methods in the “Bioinformatics” and “Phylogenetics” sections have been revised to specify our methods in greater detail.

3. Do you mean some of the assembled genomes only cover 65% of the reference genome, or reads mapping rate is 65%? I prefer only assembled genome sequences at complete and near complete to be used for subsequent analyses to avoid loss of data and inaccurate assignment of lineage and interpretation of prevalence of strains.

Thank you for identifying this issue for clarification. Our statement that “a handful of samples reporting read mapping rates below 60%...” refers to the percentage of total raw reads, with the excluded 40% of reads being removed due to human host removal. Minimum linear genome coverage achieved in this analysis was 79%, and only samples with 95% or greater linear coverage were included in the phylogenetic analysis. This section has been revised to more clearly describe this. The “Bioinformatics” section now also clarifies that only samples with >95% linear coverage was used.

---

## [Decision Letter · Decision Letter 1]

13 Aug 2025

Dear Dr. Koehler,

We look forward to receiving your revised manuscript.

Kind regards,

Nihad A.M Al-Rashedi

Academic Editor

PLOS ONE

Journal Requirements:

Reviewers' comments:

Reviewer's Responses to Questions

**Comments to the Author**

Reviewer #1: (No Response)

2. Is the manuscript technically sound, and do the data support the conclusions?

Reviewer #1: Yes

3. Has the statistical analysis been performed appropriately and rigorously?

Reviewer #1: Yes

4. Have the authors made all data underlying the findings in their manuscript fully available?

Reviewer #1: Yes

5. Is the manuscript presented in an intelligible fashion and written in standard English?

Reviewer #1: No

Reviewer #1: Title: Sequencing and analysis of 131 SARS-CoV-2 samples in previously sampled and unsampled regions of Jordan from 2020 to 2023

Overview

This manuscript presents a genomic analysis of SARS-CoV-2 samples collected across various regions in Jordan over three years. The study aims to contribute to the limited genomic data available from Jordan, particularly from under-sampled regions, and to demonstrate the utility of the EDGE COVID-19 bioinformatics platform in facilitating rapid analysis and sequencing efforts in countries developing their next-generation sequencing (NGS) capabilities.

Assessment of Incorporation of Reviewers' Comments

The authors have made significant efforts to address the reviewers' comments:

1. Acknowledgment of Sample Size Limitation (Reviewer #1 & #2):

o The authors have revised the Abstract, Introduction, and Discussion sections to acknowledge the limitation of the relatively small sample size and discuss its impact on the conclusions.

2. Expansion of Phylogenetic Analysis (Reviewer #1):

o An expanded paragraph in the Discussion now more thoroughly compares the findings to existing global and Jordanian COVID-19 pandemic phylogenetic reports.

3. Caution in Statistical Analysis (Reviewer #1):

o The authors have removed the statistical analysis limited by small sample sizes to avoid overstating conclusions. They still describe vaccination patterns qualitatively and have retained the logistic regression of SARI symptoms and vaccine type.

4. Integration of EDGE-COVID-19 Platform Discussion (Reviewer #1):

o The manuscript has been revised to better contextualize the EDGE platform in relation to the development of new NGS capabilities, integrating this discussion with the main findings.

5. Relocation of Background Information (Reviewer #1):

o The background on Jordan's healthcare system has been moved from the Introduction to the end of the Discussion.

6. Details on Quality Control Criteria (Reviewer #1):

o The Phylogenetics section now includes more details on the quality control criteria used to exclude samples from phylogenetic analysis.

7. Revised Figures for Clarity (Reviewer #1):

o Figure 1.C has been revised to use a more color-blind friendly palette.

8. Reduced Emphasis on Collaboration (Reviewer #2):

o The emphasis on collaboration has been significantly reduced, focusing more on the results and discussion.

9. Inclusion of GISAID IDs (Reviewer #2):

o The Bioinformatics section has been revised to describe how the uploaded and accepted sequences may be accessed on GISAID.

10. Clarification of Data Analysis Methods (Reviewer #2):

o The Bioinformatics and Phylogenetics sections have been revised for greater specificity, clarifying whether the sequences were mapped to the reference genome or assembled de novo.

11. Clarification on Genome Coverage and Mapping Rates (Reviewer #2):

o The authors have clarified that the mapping rates refer to the percentage of total raw reads, and only samples with >95% linear coverage were included in the phylogenetic analysis.

Overall, the authors have adequately incorporated the changes suggested by the reviewers.

Critical Review

Major Comments

1. Contribution to Genomic Surveillance:

o Strength: The study provides valuable genomic data from Jordan, especially from under-sampled regions, contributing to the global understanding of SARS-CoV-2 evolution.

o Suggestion: The manuscript could further highlight how these new sequences fill specific gaps in the global databases and their potential impact on public health strategies in Jordan.

2. Depth of Phylogenetic Analysis:

o Strength: The authors have expanded their phylogenetic analysis to compare findings with global and regional trends.

o Suggestion: A more detailed interpretation of the phylogenetic tree could be provided. Discussing possible transmission pathways, introductions of variants, and the implications for local transmission dynamics would strengthen the analysis.

3. Integration of the EDGE Platform Discussion:

o Strength: The revised manuscript better integrates the discussion of the EDGE COVID-19 platform.

o Suggestion: Providing specific examples of how the platform facilitated the research and any challenges faced would offer practical insights into its utility in resource-limited settings.

4. Analysis of Symptoms and Vaccination Status:

o Strength: The authors appropriately removed statistical analyses limited by small sample sizes.

o Suggestion: The qualitative descriptions could be enhanced by comparing them with national vaccination campaigns and existing literature to provide context and possible explanations for observed patterns.

5. Clarity and Organization:

o A thorough proofreading is necessary to improve clarity and readability.

**Do you want your identity to be public for this peer review?** For information about this choice, including consent withdrawal, please see our Privacy Policy

Reviewer #1: No

---

## [Author Response · Author response to Decision Letter 2]

16 Sep 2025

1. Contribution to Genomic Surveillance:

• Strength: The study provides valuable genomic data from Jordan, especially from under-sampled regions, contributing to the global understanding of SARS-CoV-2 evolution.

• Suggestion: The manuscript could further highlight how these new sequences fill specific gaps in the global databases and their potential impact on public health strategies in Jordan.

Thank you for suggesting that we further highlight the gaps that our sequences may fill. Efforts to combat future pandemics are informed by studying how we responded to recent ones like Covid-19. To effectively study our response to SARS-CoV-2, we need comprehensive databases and meta data (such as government response information) to draw conclusions from. A primary benefit of our sequencing is the expansion of timepoints and previously unreported strain identifications in Jordanian cities with few existing samples. While our samples are not enough to be able to broadly inform public health strategies, it will lend additional statistical power to any future global or regional analyses that include Jordan. One simple health strategy revealed by our analysis is the benefit of even small (131 samples) sequencing efforts that can reveal previously unreported strain information. This information has been incorporated into the discussion section at lines 340 - 348.

2. Depth of Phylogenetic Analysis:

• Strength: The authors have expanded their phylogenetic analysis to compare findings with global and regional trends.

• Suggestion: A more detailed interpretation of the phylogenetic tree could be provided. Discussing possible transmission pathways, introductions of variants, and the implications for local transmission dynamics would strengthen the analysis.

Thank you for requesting a more detailed interpretation of the phylogenetic analysis. While we lack much sequencing data from the important early years of the pandemic (2020, 2021), we can further interpret strain presence and distribution in 2022 and 2023 and how that may be explained by events within the country. This interpretation has been added throughout the discussion section at lines 426 – 430, 434 – 437, 440 – 450, 488 – 490.

3. Integration of the EDGE Platform Discussion:

• Strength: The revised manuscript better integrates the discussion of the EDGE COVID-19 platform.

• Suggestion: Providing specific examples of how the platform facilitated the research and any challenges faced would offer practical insights into its utility in resource-limited settings.

Thank you for suggesting that we provide examples of how EDGE COVID-19 facilitates research in limited settings. Bioinformatics analysis is often a bottleneck in genomic science as it requires intensive computation, which we did not readily have access to through our own servers. EDGE removed this bottleneck by providing free computational resources. Another strength of using EDGE COVID-19 was that it provided a suite of software specifically tailored for SARS-CoV-2. This specificity made it easier to understand and use the website for our analysis. The above information has been incorporated into our discussion of EDGE COVID-19 at lines 462 – 472.

4. Analysis of Symptoms and Vaccination Status:

• Strength: The authors appropriately removed statistical analyses limited by small sample sizes.

• Suggestion: The qualitative descriptions could be enhanced by comparing them with national vaccination campaigns and existing literature to provide context and possible explanations for observed patterns.

Thank you for identifying that qualitative descriptions could be enhanced by caparisons to national trends and existing literature. Information on Jordan’s national vaccination campaign, initiated by the Ministry of Health in mid-December 2020, has been incorporated into the discussion at lines 395 – 398, 413 - 418. Additionally, discussion of qualitative patient patterns has been supplemented with additional references to existing literature.

5. Clarity and Organization:

• A thorough proofreading is necessary to improve clarity and readability.

Thank you for this suggestion, we have reviewed the manuscript and have improved clarity and readability throughout.

---

## [Editor Report · Decision Letter 2]

6 Oct 2025

Sequencing and analysis of 131 SARS-CoV-2 isolates in previously sampled and unsampled regions of Jordan from 2020 to 2023

PONE-D-24-19649R2

Dear Dr. Koehler,

We’re pleased to inform you that your manuscript has been judged scientifically suitable for publication and will be formally accepted for publication once it meets all outstanding technical requirements.

Kind regards,

Nihad A.M Al-Rashedi

Academic Editor

PLOS ONE
---

## [Editor Report · Acceptance letter]

PONE-D-24-19649R2

PLOS ONE

Dear Dr. Koehler,

I'm pleased to inform you that your manuscript has been deemed suitable for publication in PLOS ONE. Congratulations! Your manuscript is now being handed over to our production team.

Kind regards,

on behalf of

Dr. Nihad A.M Al-Rashedi

Academic Editor

PLOS ONE